# The Therapeutic Potential of Physical Exercise in Cancer: The Role of Chemokines

**DOI:** 10.3390/ijms252413740

**Published:** 2024-12-23

**Authors:** Glenda B. B. Buzaglo, Guilherme D. Telles, Rafaela B. Araújo, Gilmar D. S. Junior, Olivia M. Ruberti, Marina L. V. Ferreira, Sophie F. M. Derchain, Felipe C. Vechin, Miguel S. Conceição

**Affiliations:** 1Health Sciences Postgraduate Program, São Francisco University, Av. São Francisco de Assis, 218, Bragança Paulista, Sao Paulo 12916-900, Brazil; glenda.buzaglo@gmail.com (G.B.B.B.); rafaelabertinii@gmail.com (R.B.A.); gilmarjr.04@gmail.com (G.D.S.J.); oliviaruberti@gmail.com (O.M.R.); marina.lvferreira@gmail.com (M.L.V.F.); 2School of Physical Education and Sport, University of Sao Paulo, Sao Paulo 05508-030, Brazil; guilherme.telles@usp.br (G.D.T.); felipevechin@usp.br (F.C.V.); 3Department of Obstetrics and Gynecology, Faculty of Medical Sciences, University of Campinas, Campinas, Sao Paulo 13083-881, Brazil; derchain@fcm.unicamp.br

**Keywords:** cancer, chemokines, chemokine receptors, physical exercise

## Abstract

The global increase in cancer cases and mortality has been associated with inflammatory processes, in which chemokines play crucial roles. These molecules, a subfamily of cytokines, are essential for the migration, adhesion, interaction, and positioning of immune cells throughout the body. Chemokines primarily originate in response to pathogenic stimuli and inflammatory cytokines. They are expressed by lymphocytes in the bloodstream and are divided into four classes (CC, CXC, XC, and CX3C), playing multifaceted roles in the tumor environment (TME). In the TME, chemokines regulate immune behavior by recruiting cells such as tumor-associated macrophages (TAMs) and myeloid-derived suppressor cells (MDSCs), which promote tumor survival. Additionally, they directly influence tumor behavior, promoting pathological angiogenesis, invasion, and metastasis. On the other hand, chemokines can also induce antitumor responses by mobilizing CD8+ T cells and natural killer (NK) cells to the tumor, reducing pro-inflammatory chemokines and enhancing essential antitumor responses. Given the complex interaction between chemokines, the immune system, angiogenic factors, and metastasis, it becomes evident how important it is to target these pathways in therapeutic interventions to counteract cancer progression. In this context, physical exercise emerges as a promising strategy due to its role modulating the expression of anti-inflammatory chemokines and enhancing the antitumor response. Aerobic and resistance exercises have been associated with a beneficial inflammatory profile in cancer, increased infiltration of CD8+ T cells in the TME, and improvement of intratumoral vasculature. This creates an environment less favorable to tumor growth and supports the circulation of antitumor immune cells and chemokines. Therefore, understanding the impact of exercise on the expression of chemokines can provide valuable insights for therapeutic interventions in cancer treatment and prevention.

## 1. Introduction

Despite its multifactorial origin, cancer is recognized as a chronic degenerative pathology often triggered by an inflammatory process [1,2,3,4]. This inflammation commonly arises from physical inactivity, obesity, and other inflammatory inducers [4,5]. Numerous agents, including cytokines, growth factors, and especially chemokines, contribute to the inflammatory processes associated with tumorigenesis (i.e., the transformation of normal cells to a neoplastic state) [2,3]. Chemokines comprise a small family pivotal to the regulation of the immune system, inflammation, and tumor behavior [6,7] that have been partially accountable for cancer development [8,9,10].

In the context of tumor behavior regulation, chemokines can create either a pro-tumorigenic or an anti-tumorigenic environment [6,8,9,10]. Their effects are determined by binding to specific receptors, which elicit various mechanisms [8,9,10,11]. Regarding the pro-tumorigenic role, studies have shown that the expression of chemokines such as CCL2 and CCL3 can recruit tumor-associated macrophages (TAMs) of the M2 subtype to the tumor microenvironment (TME), thereby enhancing the progression of ductal carcinoma in breast tissue [12,13,14,15]. In turn, M2 subtype macrophages facilitate the development of inflammation due to their pro-tumorigenic, angiogenic, and immunosuppressive actions, favoring tumor growth and invasion [15].

Chemokines can also directly contribute to pathological angiogenesis in cancer by binding to chemokine receptors expressed on endothelial cells and/or indirectly through the recruitment of progenitor and inflammatory cells [16]. This modulation of intratumoral function regulates cancer progression. The vessels produced through pathological angiogenesis exhibit abnormal structure, tumor behavior, and functions; they are heterogeneous, rough, and have an irregular vascular lumen, compromising intratumor blood perfusion [16]. This insufficient blood perfusion within the tumor characterizes a hypoxic and acidic tumor microenvironment, which supports malignancy and increases the aggressiveness of cancer cells [17]. The expression of hypoxia-related chemokines, such as CXCL12 and CXCL8, along with pro-angiogenic factors such as vascular endothelial growth factor (VEGF), produced and expressed simultaneously by tumor cells and the breast stroma, promotes the formation of abnormal vessels in tumors. This cooperation effectively drives different stages of pathological angiogenesis, which is essential for tumor maturation [16,18].

The expression of certain chemokines has also been identified as a determining factor in the metastasis process. Chemokines promote the targeted migration of cancer cells to sites of infection through the interaction between a chemokine and its receptor [19,20]. For instance, Muller et al. (2001) [20] demonstrated that breast cancer cells with active expression of the CXCR4 receptor induce actin polymerization, forming pseudopods, which are temporary projections of the cell wall. These pseudopods enable the guided migration and invasion of breast cancer cells towards organs expressing its ligand, chemokines such as CXCL12, thereby facilitating breast cancer metastasis. The intricate interaction between chemokines, tumor-associated cells, angiogenic factors, and metastasis underscores the importance of targeting these pathways with therapeutic interventions aimed at preventing tumor progression and improving patient outcomes [8,9,10].

Accordingly, specific chemokines, such as CXCR3 and its ligands, have been shown to directly mediate antitumor immunity, thereby inhibiting tumor development and increasing survival [21]. Tumor immunity can be reprogrammed through the mobilization and infiltration of various subfamilies of immune cells, such as CD8+ T cells and Natural Killer (NK) cells, into tumor cells and the TME [21,22]. Additionally, immune cells attracted by chemokines, such as CXCR3 and CCR5/4, can induce apoptosis in tumor cells through the secretion of effector cytokines, such as granzyme B, which exert a cytotoxic effect and elicit an antitumoral response [23,24]. Considering the dual role of chemokines in cancer, it is plausible to explore strategies that mitigate inflammation, thereby reducing the expression of chemokines associated with tumor development while enhancing the expression of antitumoral chemokines.

With ongoing investigations aimed at mitigating tumor development, improving prognoses, and reducing adverse effects from anti-cancer treatments, physical exercise emerges as a promising strategy, particularly due to its anti-inflammatory role [25,26,27]. Evidence suggests that exercise induces interferon-gamma (IFNy) expression, leading to increased systemic levels of the anti-inflammatory chemokines CXCL9/11-CXCR3. This process facilitates the infiltration of NK cells and CD8+ T cells into the TME, thereby enhancing anti-tumoral responses and preventing pro-tumor immune cell infiltration [28,29,30]. Moreover, physical exercise promotes physiologic angiogenesis in the TME, remodeling vascular structures, improving perfusion, and reducing intratumor hypoxia [31,32,33]. Enhanced physiologic angiogenesis facilitates the circulation and infiltration of cytotoxic immune cells such as CD8+ T cells, NK cells, and type 1 macrophages into the tumor [33,34,35]. This process is mediated by the mobilization of chemokine signaling pathways, including CXCL9/11-CXCR3 [30]. Furthermore, by reducing intratumor hypoxia and enhancing blood flow, physiologic angiogenesis decreases the production of hypoxia-inducible factor 1-alpha (HIF-1α) in the TME [36,37]. Consequently, this downregulates the expression of various mediators of pathological angiogenesis that support tumor survival and proliferation, such as chemokine CXCL12 [16,37,38]. Indeed, reducing intratumor hypoxia may facilitate and enhance anti-cancer activities [37]. Therefore, physical exercise can modulate several mechanisms orchestrated by chemokines, leading to less aggressive TME.

While the impact of physical exercise on the inflammatory process is well recognized, its specific effects on chemokines and subsequent events require comprehensive discussion to advance knowledge in the field and guide future studies. Investigating how physical exercise influences chemokine expression and their downstream effects could provide valuable insights into the mechanisms by which exercise exerts its beneficial effects on health and the modulation of cancer-related inflammation. This understanding could pave the way for tailored exercise interventions as adjunctive therapies in cancer prevention and treatment. Therefore, the present study aimed to discuss the therapeutic potential of physical exercise in cancer through the modulation of chemokines expression receptors and their link to cancer development, which are closely linked to tumor development, poor prognosis, and mortality.

## 2. The Role of Chemokine Signaling Axes in Cancer Progression

Chemokines are a subfamily of cytokines, known as “chemostatic cytokines”, responsible for the migration, adhesion, interaction, and positioning of immune system cells in the body [39,40]. Usually triggered by pathogenic stimuli, growth factors, and inflammatory cytokines such as tumor necrosis factor (TNF-α), interleukin 1 (IL-1), and IFN-*γ* and expressed by lymphocytes [10,39] in the bloodstream, chemokines are low molecular weight (8–10 kDa) molecules divided into four classes depending on the location of the cysteine (C) residues in their protein sequence: CC, CXC, XC, and CX3C [39,40].

Functionally, chemokines are classified as inflammatory when induced by an immune response at an infection/lesion site [6,41] and homeostatic (i.e., have the function of maintaining homeostasis) when involved in the control of cell migration during the development or maintenance of tissues throughout the body [41]. In addition to their essential roles in facilitating the migration of immune cells within the human body, chemokines and their receptors play multifaceted roles in cancer development [8,10]. Chemokines seem to modulate the interaction between the host and TME [7,8,10]. Particularly in this case, two families of chemokines stand out predominantly: the CC and CXC family [42].

In turn, the CXC chemokine family is determined by the absence or presence of a Glu-Leu-Arg sequence (ELR− or ELR+, respectively) [43]. While ELR− corresponds to the chemokines CXCL4, CXCL9, CXCL10, CXCL11, CXCL12, CXCL13, CXCL14, and CXCL16, the ELR+ members correspond to the chemokines CXCL1, CXCL2, CXCL3, CXCL5, CXCL6, CXCL7, CXCL8, and CXCL17 [10]. Accordingly, the receptors ELR− chemokines are ligands for CXCR3, CXCR4, CXCR5, CXCR6, or CXCR7, and ELR+ are ligands for CXCR1 and/or CXCR2 [10,44]. Additionally, the subfamily of CXC chemokines has dual function in tumor dynamics, acting in tumor promotion or suppression, depending on the context of cell signaling. ELR+ CXCLs/CXCR1/2 signaling promotes tumor progression through multiple signaling pathways [10]. When expressed by cancer-associated cells such as TAMs within the TME, they specifically activate signaling pathways that mediate tumor cell proliferation and metastasis [45,46,47] in addition to recruiting cells associated with cancer-increasing tumor malignancy [10] such as the mitogen-activated protein kinase (MAPK), the 3-kinase phosphoinositide (PI3K)/protein kinase B (AKT) [47,48], the signal transducers and activators of transcription (STAT3) [46], and the nuclear kappa B (NF-κB) [45]. Contrarily, ELR-CXCLs/CXCR3/4/5/6/7 signaling has mainly tumor suppressive effects. For example, CXCR3, when expressed by CD8+ T cells, allows their ligands CXCL9 and CXCL10 to guide the migration of CD8+ T cells to tumor tissue, resulting in an anti-tumor effect. However, despite the antitumor effect, it is noteworthy that in some cases the ELR− ligands have pro-tumorigenic properties, such as ELR-CXCL12/CXCR4 [20], ELR-CXCL11/12-CXCR7 [44], and ELR-CXCL13/CXCR5 [49], increasing angiogenic and metastatic activities in the tumor [50,51].

Therefore, chemokines are essential to activate cancer signaling pathways, enable immune escape mechanisms employed by cancer cells, and facilitate the recruitment of tumor-associated cells, such as tumor-associated TANs, TAMs, MDSCs, and T_reg_ [8,9,10], contributing to the development and immunosuppression in TME. Furthermore, chemokines enhance pathological angiogenesis, promote cancer cell proliferation, and facilitate metastasis [16,20,52]. Additionally, they also act in cancer suppression by recruiting immunological cells such as NK cells, CD8+ T cells, macrophages M1, T helper (Th1), and B cells [53,54] towards TME. This complex interplay underscores the significance of targeting chemokine signaling pathways in cancer therapy to disrupt these processes and potentially improve treatment outcomes. Table 1 and Table 2 summarize the chemokines, respective receptors, and effects in the tumor.

## 3. Chemokines and Immune Cells Interaction in Cancer Development and Progression

The recruitment of specific immune cells, such as TANs, TAMs, and T_reg_ cells, can increase tumor promotion [55]. The recruitment of tumor-associated macrophages (TAMs) to tumor sites is mediated by chemokines such as CCL2, CCL5, CCL7, CXCL1, CXCL2, and CXCL4 [8,55], which promote tumor survival. In addition to this mobilization, chemokines such as CCL2, CCL3, CCL4, and colony-stimulating factor 1 (CSF-1), which are already expressed by cancer cells at the tumor site, drive TAM differentiation and polarization into either classically activated M1 macrophages or immunosuppressive M2 macrophages within the TME [51]. These macrophage subtypes play crucial roles in the interaction between cancer and inflammation. Evidence indicates that the high concentration of polarized M2 (i.e., M2 and M1 converted to M2) macrophages in TME allows the growth/development of tumors, as in gastric cancer, due to its pro-tumorigenic action [56]. Additionally, the M2 phenotype is closely associated with immunosuppression and the promotion of pathological angiogenesis [51,57] through the secretion of pro-angiogenic factors such as VEGF and fibroblast growth factor (FGF), which play a crucial role in enhancing pathological angiogenesis and tumor proliferation [50,58]. TAMs and TANs can be recruited into the tumor microenvironment by chemokines produced by cancer cells, such as the chemokines signaling axis CXCL1, CXCL2, CXCL5, CXCL8–CXCR2, and CXCR1 [59,60]. Also, TAMs and TANs can differentiate and be polarized by the action of cytokines such as interferon *β* (IFN-*β*) and transforming growth factor *β* (TGF-*β*), resulting in neutrophils of the N1 (antitumor) and N2 (protumor) phenotypes, resulting in the N2 phenotype, which favors tumor proliferation and metastasis [61,62], further strengthening cancer development.

Crucially, MDSCs are recruited to tumor sites via various chemokine axes, such as CCL8/CCR2, CCL5/CCR5, CXCL5/CXCR2, and CXCL12/CXCR4 [54,63,64]. After chemokine-mediated infiltration of MDSCs in TME, infiltrated MDSCs increase the activity of signal transducer and activator of transcription 1 (STAT1), reducing the levels of reactive oxygen species (ROS) and increasing the levels of inducible nitric oxide synthase (iNOS), nitric oxide (NO), and arginase-1, inhibiting CD8+ T cells in the tumor [55,64]. In addition, T_regs_ are also involved in reducing the immunological activity through the suppression mechanism of tumor-specific T cells [6,65]. T_regs_ are influenced by chemotaxis through chemokine signaling pathways, such as CCL20/CCR6 and CXCL12/CCR4, which guide their migration towards the TME [66,67,68]. This recruitment contributes to rendering the TME more intolerant to the effects of anticancer agents and facilitates the suppression of antitumor cells, including CD8+ T cells [54,65], favoring tumor proliferation and survival.

Chemokine axes are also essential in recruiting immune cells with an anti-tumoral effect [54,55]. Once expressed and activated, chemokines CCL5/3/19/20, CXC21, CXCL9/10/12/13, and CX3CL, and their binders CCR4/5/7, CXCR3/4/6, and CX31 [21,22,63,69,70], can induce Nk, CD8+ T, T help (Th1), and B cells [53,54], chemotaxis to the TME triggering a direct antitumor effect. For instance, CCR4/5 and CXCR3, expressed by CD8+ T cells, allow their ligands CCL5, CXCL9, and CXCL10 to guide the migration of CD8+ T cells into tumor tissue. In response to this migration, CD8+ T cells enable apoptosis in tumor cells through the secretion of effector cytokines such as granzyme B (cytotoxic effect), resulting in an effective antitumor effect [23,24]. Considering these antitumoral effects, Peng et al. (2019) performed the immunohistochemical analysis of tumor samples from 122 women diagnosed with breast cancer in an attempt to detect the expression levels and localization of FoxP3 and CD8 in invasive ductal breast carcinoma. The authors also investigated the correlations between FoxP3 + (T_regs_) regulatory T cells and CD8 + cytotoxic T lymphocytes (CTLs). The study revealed a higher expression of FoxP3 in the tumor bed, contrasting with the predominant expression of CD8 in the stroma of breast tissue. This distribution suggests that CTL infiltration into the stroma is associated with a favorable prognosis for breast cancer patients [71]. Similarly, chemokine axes such as CCR5-CCL3/CCL5 [22] and CXCR3-CXCL10 [21] facilitate the NK cell-mediated cytotoxicity-inducing antitumor response [72,73].

The CXCR3-CXCL9/CXCL10 chemokine axis, mobilized by Th1 cells [74,75], can also participate in antitumor responses remodeling tumor immunity through the production of interferon gamma (IFN-γ). It has been shown that IFN-*γ* interrupts cell proliferation, promotes apoptosis and reduction of angiogenesis, decreases the chance of metastases, and improves antitumor immune responses [76]. In addition, IFN-*γ*, also produced by Th1 cells, has been shown to enhance the direct cytotoxic activity of macrophages against cancer cells, stimulating macrophages to secrete chemokines such as CXCL9 and CXCL10, which may contribute to an anti-tumor environment, potentially preventing cancer progression rather than promoting it [77]. Additionally, chemokine axes such as CXCL12/CXCR4, CXCL13/CXCR5, CCL19, CCL21/CCR7, and CCL20/CCR6 are associated with mobilization and infiltration of B cells in tumors [70,78]. B cells are intricately involved in combating tumors through various mechanisms, presenting also anti-tumor functionality. They play a direct role in inducing tumor cell death and producing antibodies targeting tumor antigens [70,77,79]. Also, B cells act as antigen-presenting cells (APCs), crucial for activating T cells and promoting the formation of memory T cells. By facilitating the immune responses of cytotoxic CD8+ T cells, B cells contribute significantly to antitumor immunity [80,81,82]. However, B cells also provide pro-tumoral effects by activating STAT3, enabling pathological angiogenesis and facilitating tumor progression [83]. Despite the dual role of B cells in some types of cancer (e.g., non-small cell lung), their increase is correlated to long-term survival, both in patients in the early stage and advanced stage of chemotherapy treatment. Chemokines indeed play a crucial role in recruiting immune cells in both pro-inflammatory and anti-inflammatory processes. They act as molecular messengers, guiding immune cells to sites of inflammation or tissue damage, where they participate in immune responses aimed at combating pathogens or resolving inflammation.

### 3.1. Chemokines in Tumor Angiogenesis

Chemokines can also influence various other biological processes beyond their role in immune cell recruitment, including pathological angiogenesis. Angiogenesis is the process of forming new blood vessels from existing ones, and it is essential for supplying oxygen and nutrients to tissues, including tumors. In general, tumor progression is accompanied by blood vessel growth, which is essential to tumor maturation [16]. However, the tumor vasculature is abnormal and immature (i.e., they are heterogeneous, rough, and have an irregular vascular lumen), a process known as pathological angiogenesis. Pathological angiogenesis prevents the transport of oxygen and the removal of residues in the TME, as in normal blood vessels [16,84,85]. Concomitantly, the high production of interstitial fluid (IF) induces the expression of the production of HIF1α and consequent intratumorally hypoxia [16,82]. Consequently, a continuous cycle is generated, which leads to tumor growth. Together, these processes result in the inhibition of blood perfusion, leading to lower anticancer responses and increasing pathological angiogenesis [16]. Pathological angiogenesis can occur through the upregulation of specific chemokines (e.g., CXCL8 and CXCL12) that activate pro-angiogenic mediators, such as vascular the VEGF and/or recruit progenitor endothelial cells for carcinomas (EPCs), stimulating the formation of new blood vessels and cancer progression [16,86]. ELR + CXC chemokines, including CXCL1, CXCL2, CXCL3, CXCL5, CXCL6, and CXCL7, increase angiogenesis by binding to their common CXCR2 receptor expressed by EPCs [16,87]. These axes can stimulate the proliferation, survival, and migration of EPCs, providing an increase in the formation of tumor microvessels [10,16]. In addition, chemokine CXCL12 induced by hypoxia-induced factors (e.g., HIF-1α) also has a potent angiogenic action when linked to its CXCR4 receptor, highly expressed in tumor vessels, triggering an increase in blood vessel formation [16,86]. Other members of the CCL family are also involved in this process. Lin et al. (2015) observed that chemokine CCL18, expressed by macrophages M2, promoted pathological angiogenesis and tumor progression in breast cancer, both in vitro and in vivo [88]. Collectively, the intricate roles of chemokines in cancer progression extend beyond immune cell recruitment to include significant contributions to pathological angiogenesis. The abnormal formation of blood vessels is crucial for tumor maturation and results in dysfunctional vasculature that hinders efficient blood perfusion and oxygen delivery, thereby promoting hypoxia and tumor growth. The upregulation of specific chemokines and their interactions with receptors underscore their pivotal role in driving angiogenesis and cancer progression. Understanding these mechanisms provides valuable insights into potential therapeutic targets for disrupting the pro-tumorigenic functions of chemokines, ultimately contributing to more effective cancer treatments.

### 3.2. Chemokines in Tumor Metastasis

Tumor metastasis is a consequence of tumor cell extravasation that potentiates cancer deaths. Firstly, tumor cells migrate from their original site through the bloodstream and/or lymphatic vasculature to lodge in a distant tissue to form a new colony of tumor cells [89]. Interestingly, the literature has been describing the active contribution of chemokines and their respective receptors with the new colonization of tumor cells [10,54,90], enhancing the metastasis process and potentializing cancer-induced mortality.

Some chemokines can induce deterioration of the extracellular matrix, a pathway to tumor dissemination and invasion, mainly through the action of matrix metalloproteinases (MMPs). For instance, overexpression of the chemokine CXCL13 activates the oncogenic pathways STAT3 and extracellular signal-regulated kinase ERK1/2, also inducing the expression of MMPs, such as MMP2 and MMP9, that are determinants of metastasis [91]. Yang et al. (2019) observed that expression of CXCL1 activates ERK-MMP2-MMP9 signaling pathways, stimulating cell migration and invasion and promoting metastasis in estrogen receptor-negative breast cancer [92]. In lung cancer, it was shown that the positive of the CXCL13-CXCR5 axis induces high expression of vascular adhesion molecule-1 (VCAM-1), which enhances cell migration [49] and consequently metastasis. Finally, other chemokines, such as CXCL1, are also involved in the formation of the premetastatic niche (PMN). CXCL1 recruits hematopoietic stem and progenitor cells (HSPCs), favoring the differentiation of MDSCs for the promotion and formation of PMN [93], providing growth of metastatic cells.

Indeed, chemokines play a crucial role in fostering a TME conducive to metastasis in various types of cancer, as shown in Figure 1 [7,8,9,10,11]. As integral immune system components, their involvement in cancer development and metastasis necessitates particular attention. Efforts to diminish the pro-tumoral effects of chemokines while enhancing antitumoral responses mediated by chemokine axes are imperative.

## 4. Physical Exercise: Chemokines Action Power

A specialized literature investigating the intricate interplay between chemokines, immune cells, and cancer cells offers promise for devising novel therapeutic strategies targeting chemokine signaling pathways in cancer. In this context, physical exercise has emerged as a potential adjunctive therapy in cancer management. Studies suggest that regular exercise can positively impact the immune system, bolstering antitumor immune responses and decreasing tumor progression, as shown in Figure 2 [26,30]. Through the modulation of immune function and the promotion of an unfavorable microenvironment for tumor growth, physical exercise may contribute to improved treatment outcomes and prolonged survival in cancer patients [26,27,94].

Regular exercise induces a challenge to body homeostasis [95] by eliciting adaptations to cellular, molecular, and systemic levels [27,95]. Accordingly, the literature has linked the antitumoral effects of physical exercise to a decrease in tumor risk factors, including chemokines [25,26,27,30,36,96]. Indeed, it has been shown that physical exercise induces chemokines that act as anticancer agents [30,97]. Therefore, understanding the role of chemokines expressed during or after physical exercise in tumor development and progression is crucial.

Physical exercise induces shear stress and adrenaline release, both associated with increased vigilance and regulation of the immune system [96,98,99]. Specifically, the release of adrenaline induces lymphocytosis (an increase in lymphocytes in the circulation), establishing a state of immunocompetence marked by a redistribution of immune cells throughout the body [96,99]. This facilitates cell mobilization and, consequently, the release of chemokines for the TME [100]. Accordingly, Gomes-Santos et al. (2021) investigated the effects of seven sessions of aerobic exercise (i.e., the chronic effect of exercise) on tumor-infiltrating lymphocytes (TILs). The study revealed that repeated exercise sessions lead to a notable increase in the expression of CD8+ cells and their infiltration in the TME. Additionally, it was demonstrated that repeated exercise sessions increase the expression of CXCL9/CXCL11-CXCR3, which are involved in enhancing the trafficking of CD8+ T cells to the tumor, consequently boosting antitumor activity through immunological reactivity, the direct attack on cancer cells in the TME [30]. Moreover, Esmailiyan et al. (2022) investigated the effect of accumulated eight-week noncontinuous aerobic training (i.e., chronic effect of exercise) on CCL2 and CCL5 levels and their related receptors CCR2 and CCR5 in BALB/C female mice with breast cancer. Compared with the control group (no exercise), a significant decrease in plasma levels of pro-tumorigenic chemokines CCL2 and CCL5 and their respective CCR2 receptors was observed [97]. This suggests a reduction in the recruitment of macrophages M2 towards TME as a chronic response to exercise training. This decrease corroborates with the decreased tumor volume and immunological reprogramming observed in the study [97]. It has been demonstrated that aerobic training performed during sixteen weeks by women recovering from breast cancer induced a significant reduction in the systematic concentration of TNF-1α, NK cells, and natural killer T cells (NKT) compared to the control group [101]. In addition to aerobic training, resistance training also seems efficient in chronically reducing pro-inflammatory markers and improving the immune status of women recovering from breast cancer [101,102,103,104]. Altogether, these results suggest that physical exercise might have the capacity to activate immune mechanisms, bolstering the defense system through the expression and mobilization induced by chemokines. In turn, this mechanism reduces aggressiveness within the TME, ultimately leading to a decrease in tumor size.

The efforts induced by repeated exercise sessions can also stimulate chronic physiological adaptations in the TME, such as the dilation and maturation of intratumoral vasculature. This is evidenced by increased coverage of pericytes in microvessels [105], associated with elevated body temperature during an exercise bout [95,106] The intratumoral vasculature dilation, modulated by body temperature, would also be able to increase the circulation of immune cells in the tumors during exercise in both murine and human [106,107,108,109,110]. This migration can be guided by chemokines such as CXCR3-CXCL9/CXCL10 along a chemical gradient of immunological cells, such as CD8+ T cells and NK cells [111]. Accordingly, Burd et al. (1998) noted that inducing hyperthermia in mice with xenografted breast tumors not only enlarged the diameter of intratumoral blood vessels but also enhanced tumoral infiltration of NK cells [106], contributing to a delay in tumor growth. Physiologically, the dilation and maturation of intratumoral vasculature induced by exercise that increases the antitumoral effect through chemokines response might also induce enhanced tumor perfusion with a consequent increase in oxygenation and reduction of intratumoral hypoxia chronically [31,33,36,105].

Some studies have shown that hypoxia is one of the factors associated with the suppression of antitumor immune responses [37], by directly reprogramming immune cells for the pro-tumor phenotype [112], modulating the signaling axes of chemokines [113,114], and inhibiting the cytotoxicity of immune cells [115]. Accordingly, Pedersen et al. (2016) observed that accumulated sessions of physical exercise (i.e., chronic effect) induced low levels of hypoxia and a high expression of CCL3 and CXCL10 chemokines, which are linked to the activation of chemotaxis of immune cells, such as NK in mice [116], towards the TME, to reprogram tumor immunity. Furthermore, McCullough et al. (2014) showed that only one session of physical exercise (i.e., acute effect of exercise), performed by copenhagen rats with orthotopic prostate tumors, was able to increase blood flow in the tumor by up to 200% with a significant increase in oxygen delivery. These changes were accompanied by a decrease in approximately 15% of hypoxic tumor areas [117]. In another study [118], McCullough and colleagues showed that, chronically, physical exercise training provided a greater reduction in intratumor hypoxia in rats with prostate tumors when compared to sedentary rats, suggesting that regular physical exercise can facilitate infiltration of anticancer agents such as chemokines linked to the mobilization and circulation of immune cells towards TME.

Linked to immune suppression triggered by hypoxia, lactate metabolism in the TME is also relevant [119,120,121]. An important characteristic of tumors is the Warburg effect, a phenomenon described by Otto Warburg in which cancer cells prefer to metabolize glucose through anaerobic glycolysis (i.e., lactic acid fermentation) instead of producing energy through oxidative phosphorylation [120]. The tumor glycolytic phenotype increases lactate production by cancer cells, making TME more acidic and inhibiting T cell responses, which may promote the production and greater circulation of pro-inflammatory chemokines, damaging the immune pathways that could regulate tumor growth [119]. Therefore, several authors pointed out that lactate metabolism is a determinant for malignancy, immune suppression, and cancer prognosis [119,120,122]. Since increased glucose consumption is directly linked to tumorigenesis, reversing this metabolic profile could compromise tumor growth and survival [120]. Importantly, regular physical exercise, especially high-intensity exercise, can impact tumor metabolism, inhibiting glycolysis [119,122]. Parolin et al. [123] analyzed the time course for activating glycogen phosphorylase and pyruvate dehydrogenase in human skeletal muscle during repeated bouts of maximal exercise. The authors demonstrated that with successive bouts, the ability to stimulate substrate phosphorylation through phosphocreatine hydrolysis and glycolysis decreased, shifting metabolism toward greater dependence on oxidative phosphorylation [123]. In summary, the induced acidosis during exercise leads to reduced lactate production as a result of glycogenolysis inhibition [124]. These lactate-inhibiting effects may enhance the distribution and transmigration of immune cells and anti-tumor chemokines to tumors, facilitating tumor lysis [125,126].

Associated with tumor vasculature improvements and hypoxia reduction, induced by exercise, there is also a reduction in the production of hypoxia-induced alpha-1 factor (HIF-1α). Decreases in HIF-1α induce a downregulation of the mediators of a series of genes involved in pathological angiogenesis, such as chemokine CXCL12 [37,127,128]. Not less important, physical exercise induces shear stress in the tumor vascular wall and may modulate angiogenic factors (e.g., VECF-A, VEGF-R2, CD31) and the chemokine CCL3 in endothelial cells, which are strongly related with stimulating physiologic angiogenesis stimulation [26,36,105,129,130]. Therefore, we propose that physical exercise may mitigate tumoral pathological angiogenesis, leading to reduced hypoxia and improved functionality of antitumor agents, including chemokine signaling pathways.

In summary, the interplay between chemokines, immune cells, and cancer cells offers promising therapeutic strategies. Physical exercise has emerged as a potential adjunctive therapy in cancer management. A single session of exercise induces immediate immune responses, such as the release of adrenaline and increased lymphocytosis, facilitating the mobilization of immune cells and chemokine release in the TME. Repeated exercise sessions lead to long-term adaptations, reducing pro-tumorigenic chemokines such as CCL2 and CCL5, and promoting antitumor immune responses through improved vasculature and oxygenation in the TME. Studies have shown that regular exercise enhances CD8+ T cell infiltration and reduces hypoxia, thereby mitigating tumor growth. By incorporating regular exercise into cancer treatment, patients may benefit from enhanced immune responses, reduced tumor aggressiveness, and improved treatment outcomes. It is important to note that the majority of the studies cited here investigated immunological mechanisms using cell models and animal subjects. Therefore, future research should aim to replicate these findings in studies involving cancer patients to better understand their clinical relevance.

## 5. Conclusions and Future Perspectives

Chemokine signaling pathways can either promote or inhibit tumor growth, depending on the context and stage of the carcinogenic process. When triggered by physical exercise, certain chemokines, particularly CXCL10, CXCL9, CXCL11, and their receptor CXCR3, can elicit anti-tumoral effects. The anticancer effects induced by immune modulation resulting from physical exercise represent a non-pharmacological, safe, effective, and cost-efficient strategy.

Considering that physical exercise training can reduce the likelihood of tumor development and proliferation by strengthening the immune system—especially through the action of chemokines—we strongly recommend initiating physical exercise for cancer patients immediately after diagnosis. Exercise before and during treatment can enhance the signaling of anti-cancer chemokines, increase the number of circulating lymphocytes, and improve their mobilization within the TME.

It is important to tailor exercise recommendations to individual patient limitations and medical advice. The prescription of exercise to cancer patients, both before and during treatment, can positively modulate immunological activation. This low-cost and non-invasive strategy can facilitate a better response to oncological treatments. Therefore, physical exercise serves as a powerful complementary tool to conventional cancer therapies. We also emphasize that incorporating physical exercise should be a standard consideration within hospital settings and among all multidisciplinary teams involved in the care of cancer patients.

## Figures and Tables

**Figure 1 ijms-25-13740-f001:**
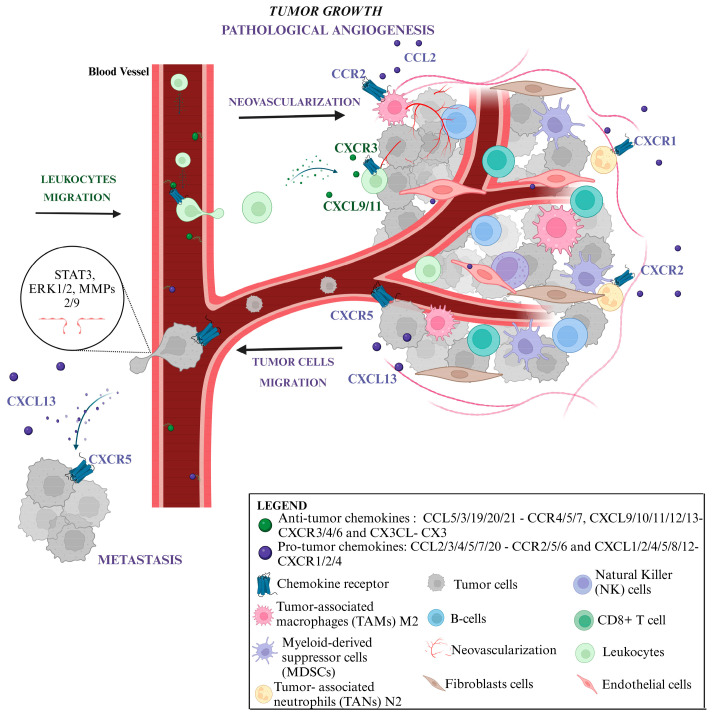
Mechanisms connecting chemokines and their receptors to tumor microenvironment. Solid tumors contain numerous types of stromal cells, such as endothelial cells and fibroblasts, which are major producers of chemokines. These chemokines play a crucial role in the regulation and migration of leukocytes in both tumor bed and TME. The circulation of leukocytes is a highly coordinated process, in which they roll along the endothelium. Chemokines bind to leukocytes through their protein-coupled receptors, resulting in a strong adhesion of leukocytes to the endothelial layer. Eventually, leukocytes overtake the endothelium and migrate towards chemokine-producing tissue or cells. Consequently, the tumor is infiltrated by inflammatory cells, including neutrophils, M2 macrophages, T lymphocytes and dendritic cells. CXC ELR+ chemokines attract tumor-associated neutrophils (N2) expressing CXCR2 and CXCR1. Similarly, chemokines of the CC subfamily attract tumor-associated macrophages (M2) expressing CCR1/2/3/5/8. In contrast, some ELR-CXC chemokines, such as CXCL9/10/11 ligands, attract CXCR3-activated T lymphocytes and NK and CD8+ T cells, which may exert antitumor (cytotoxic) activity. Regarding the production of chemokines by leukocytes, tumor cells and tumor-associated cells (Tam’s subtype M2, Tan’s subtype N2, MDCs, T_reg_) have effects on pathological angiogenesis through their angiogenic ligands (CXCL1/2/3/5/6/7/8-CXCR2; and agonist CXCR4 CXCL2; CCL2/7/8/13/16-CCR2; CCL20-CCR6; CCL1/18-CCR8; CCL27/28-CCR30; CCR3). This triggers the formation of more immature and hypoxic intratumor vessels. The formation of new blood vessels is determinant in tumorigenesis, because it maintains tumor survival and enhances its proliferation, boosting the process of metastasis. In addition, the receptor-ligand axis CXCL13–CXCR5 and CXCR4–CXCL12 is involved in the targeted migration of tumor cells to metastatic sites, through the activation of oncogenic pathways (STAT3, ERK1/2 and MMPs 2/9) that deteriorate the extracellular matrix, facilitating the escape of cancer cells. TME: tumor microenvironment; STAT3: signal transducers and activators of transcription; ERK1/2: extracellular signal regulated kinase; MMPs2/9: matrix metalloproteinases. Created with Biorender.

**Figure 2 ijms-25-13740-f002:**
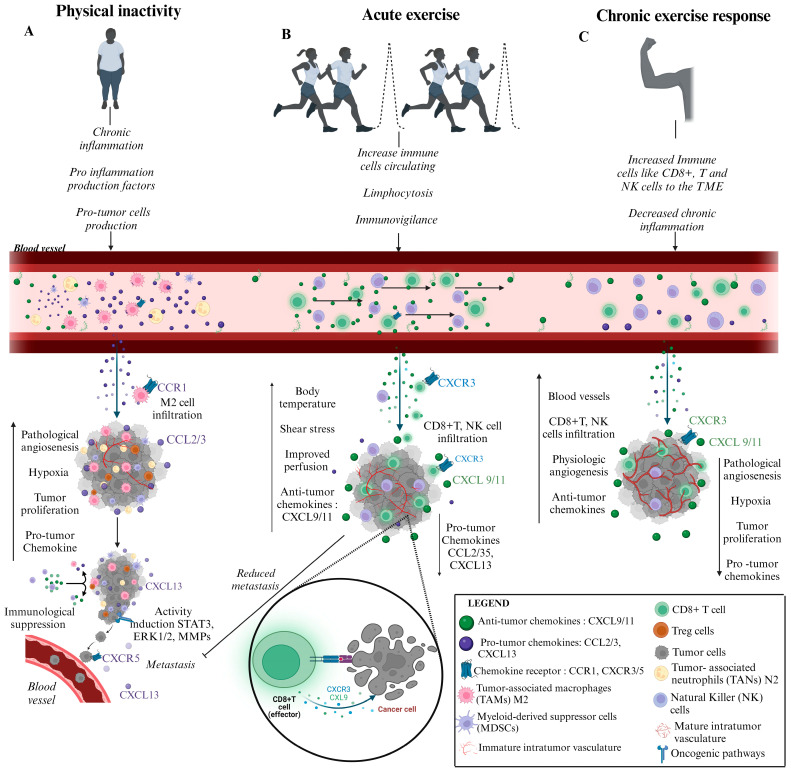
Systemic and Molecular Mechanisms Associated with the Modulation of Chemokines by the Immune System and Changes in the Tumor Microenvironment (TME). (**A**) Physical inactivity is one of the mediators of the inflammatory process. When this inflammation worsens and becomes chronic, there is an excessive production of pro-inflammatory factors, such as the chemokines CC2, CCL3-CCR1, and CXCL13-CXCR5, among others mentioned in this article. These substances aid in the production and excessive migration of pro-tumor cells such as TAMS, TANS, MDSCs, and T-reg into the bloodstream, directing them to an organ of the human body and triggering the formation of a primary tumor. As the signaling axis of chemokines CC2, CCL3-CCR1 facilitates the migration of Tams from the subtype M2 to the TME, this potentiates the inflammatory process. This same axis contributes to increased pathological angiogenesis and intratumor hypoxia, presenting an immature vasculature and low oxygen circulation. When the production and infiltration of CXCL13-CXCR5 chemokines, as well as of TAM pro-tumor cells, TANs of M2 and N2 subtypes, and especially of T reg, becomes uncontrolled, there is an enhanced immunological suppression and tumor proliferation. Additionally, the uncontrolled production of the CXCL13-CXCR5 chemokine axis activates oncogenic pathways, such as STAT3, ERK1/2, and MMPs 2/9, which induce the deterioration of the extracellular matrix. This favors the escape of tumor cells induced by chemokine receptors, such as CXCR5, into the blood circulation, determining the process of metastasis. (**B**) Single sessions of exercise promote immune regulation, such as increased circulation of immune cells due to the lymphocytosis process, which mainly affects NK cells and CD8+ T cells. This lymphocytosis is driven by physical changes in the body, such as increased blood flow, shear stress, and increased body temperature, resulting in a higher concentration of immune cells in the blood circulation. Subsequently, a transient lymphopenia occurs, in which the mobilized cells are redistributed to the infection sites, improving immunovigilance and increasing the cytotoxic activity of immune cells. This increase in the circulation of NK and T CD8+ cells stimulates the production of antitumor chemokines, such as CXCL9, CXCL11, and CXCR3 (among others already mentioned in the article), which promote the targeted migration of these cells to TME. Additionally, physical changes induced by exercise sessions (increased blood flow, shear stress in the vascular bed, and elevation of body temperature) act directly in the TME, tumor perfusion, and intratumor oxygen supply. This facilitates infiltration of anti-tumor chemokines CXCL9/11-CXCR3 and NK and T CD8+ cells, increasing their cytotoxic activity against tumor cells and reprogramming intratumor immunity. Consequently, there is a reduction in the expression of CXCL13-CXCR5 chemokines and pro-tumor cells associated with the activation of oncogenic signaling pathways linked to metastasis. (**C**) Repeated bouts of exercise lead to chronic adaptations that include systemic changes, such as improved immune function, with increased circulation of NK immune cells, CD8+ T, and anti-tumor factors. In addition, there is a reduction in chronic inflammation and changes in the tumor microenvironment triggered by physiological angiogenesis, which improves blood perfusion and maturation of tumor vasculature, reducing intratumor hypoxia. These adaptations facilitate greater infiltration of NK and T CD8+ cells and their cytotoxic action, as well as high concentrations of CXCL9/11-CXCR3 chemokines in TME, as a result of immunological reprogramming. Thus, all adaptations corroborate a reduction in tumor proliferation and a low concentration of pro-tumor chemokines. TME: tumor microenvironment; STAT3: signal transducers and activators of transcription; ERK1/2: extracellular signal regulated kinase; MMPs2/9: matrix metalloproteinases. Created with Biorender.

**Table 1 ijms-25-13740-t001:** CC chemokine receptors (CRC), with their ligands and functions in a tumor.

Receptor	Ligand	Cells Recruited for the Tme	Effect on the Tumor
CCR1	CCL2, CCL3, CCL4, CCL5, CCL7, CCL8, CCL14, CCL15, CCL16, CCL23	MDSCs, MSC, TAMs, TANs	↑↑ VEGF; Pathological angiogenesis; Assist tumor stroma remodeling; Facilitates tumor survival
CCR2	CCL2, CCL7, CCL8, CCL13, CCL16	MDSCs, MSC, TAMs, TANs, T_reg_ cells	Pathological angiogenesis dependent on TAM; Survival and resistance of tumor cells
CCR3	CCL5, CCL7, CCL8, CCL11, CCL13, CCL14, CCL15, CCL24, CCL26, CCL28	Eosinophils, TAMs	Pathological angiogenesis
CCR4	CCL2, CCL17, CCL22	TILs, Th17 cells, T_reg_ cells	Antitumor immune suppression
CCR5	CCL3, CCL4, CCL5, CCL7, CCL11, CCL14, CCL16	CAF, TILs, MDSCs, TAMs, T_reg_ cells	↑↑ VEGF; Immune infiltration; Metastatic dissemination.
CCR6	CCL20	TAMs, Th17, T_reg_ cells	Pathological angiogenesis
CCR7	CCL19, CCL21	T_reg_ cells	↑↑ VEGF VEGF-A, VEGF-C, and VEGF-D; Pathological angiogenesis; Lymph node metastasis
CCR8	CCL1, CCL16, CCL18	TAMs, T_reg_ cells	Pathological angiogenesis; Immune evasion; Immunological suppression; Tumor survival; Lymph node metastasis.
CCR9	CCL25	-	Metastatic dissemination
CCR10	CCL27, CCL28	T_reg_ cells	Migration of tumor cells to the skin and mucosa

↑↑— Increased expression; MDSCs—myeloid-derived suppressor cells; mesenchymal MSC—stem cells; TAMs—tumor-associated macrophages; TANs—tumor-associated neutrophils; Th17—T helper 17; TILs—tumor-infiltrating anticancer lymphocytes; T_reg_—regulatory cells; VEGF—vascular endothelial growth factor. CAF—fibroblasts associated with cancer.

**Table 2 ijms-25-13740-t002:** CXC chemokine receptors (CXCR), with their ligands and functions in a tumor.

Eceptor	Ligand	Cells Recruited for the Tme	Effect on the Tumor
CXCR1	CXCL5, CXCL6, CXCL8,	MDSCs, TAMs, TANs, NK, CD8+ T cells	Proliferation; Survival; Pathological angiogenesis; Immunosuppressive; Metastasis; Chemoresistance
CXCR2	CXCL1, CXCL2, CXCL3, CXCL5, CXCL6, CXCL7, CXCL8	MDSC, TAMs, TAN, NK, CD8+ T cells	Tumor proliferation; Pathological angiogenesis; Migration/invasion; Immunosuppression; Chemoresistance
CXCR3	CXCL4, CXCL9, CXCL10, CXCL11, CXCL13	CD8+ T cells activated, CD4+ T cells activated, Th1 cells, NK, NKT, B cells	Inhibit pathological angiogenesis; Increases antitumor immunity; Promote angiogenesis in some types of cancer (colorectal cancer); Promotes proliferation, migration, metastasis, and tumor recurrence (see Penis Cancer, Gastric Cancer, Leukemia)
CXCR4	CXCL12	T_reg_, B cells, TANs, TAMs, MDSCs	Tumor growth, Pathological angiogenesis, Metastasis; cell survival and proliferation
CXCR5	CXCL13	CD8+ T cells, Th17 cells, T_reg_ cells, B cells	Proliferation, Migration, Lymph node metastasis, TME modulation
CXCR7	CXCL12	CAF, TILs, MDSCs, TAMs, T_reg_ cells	Pathological angiogenesis; Invasion; Metastasis; Survival; Proliferation
CXCR6	CXCL16	NK e CD4+ T e CD8+ activated cells	Increase the anti-tumor immunity

MDSCs—myeloid-derived suppressor cells; mesenchymal MSC—stem cells; TAMs—tumor-associated macrophages; TANs—tumor-associated neutrophils; Th17—T helper 17; TILs—tumor-infiltrating anticancer lymphocytes; T reg—regulatory cells; VEGF—vascular endothelial growth factor. CAF—fibroblasts associated with cancer; NK—natural killer cells; NKT-T—natural killer cells; T CD4+ cells activated—T CD4+ cells; T CD8+ activated—T CD8+ cells; Th1—T Helpe 1 cell; B cell.

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
