# Peer review of "The Therapeutic Potential of Physical Exercise in Cancer: The Role of Chemokines"

_ijms, 2024, doi:10.3390/ijms252413740_

Round 1
Reviewer 1 Report
Comments and Suggestions for Authors
The manuscript by Buzaglo et al., “The therapeutic potential of physical exercise in cancer: the role of chemokines”, is a review focused on the actions of chemokines and their role – positive and negative – in governing the interactions between the immune system and cancer. As reflected in the title, with an emphasis on how exercise influence chemokine axes, and how that could be exploited in cancer therapies.
The review is very timely, and well organized and written. Below some points raised.
I raise the following concerns.
The authors have chosen the term “chronic” for repeated bouts of (acute) exercise. I am not sure that is the most appropriate term. I think it makes more sense to speak about repeated (for whatever period relevant) bouts or regular (programs of) acute exercise. Chronic exercise, e.g., running for really long time, e.g., marathon and beyond could perhaps signify as chronic, but would not be likely to set a similar anti-tumor state of the immune system?!?
The term “tumor behaviour” (first used line 48) is odd, I think. Would “tumor composition” or “cellular and molecular composition of the tumor” be more appropriate?
The paragraph starting line 258, is set of in the first line as if it will be about Th1 cells, but in fact its more about B cells.
The review is mainly focused on data from mouse tumor models, reflecting that the research field of exercise immunology/oncology is understudied in the human setting. Nonetheless, some data are available that could potentially be used to strengthen the notion that the mouse studies can be translated to the human setting. Also, clinical studies are in progress combining exercise with immunotherapy in cancer, coupled with biological monitoring (PMID: 35247994).
Minor concerns
At least on my print and screen the quality/resolution of fig. 1 is low.
Several place there is a lack of space between last letter and reference, i.e., “…..immune[“, which should be “….immune [“
When authors are referred to in the text it should be e.g., “Pedersen et al.” and not “Pedersen L. et al.”)
I suggest to replace “induction” (line 86) with “mobilization”.
I suggest to replace “immunological cells” (line 89) with “immune cells”.
I suggest to replace “induced” (line 89) with “attracted”.
I suggest clarifying lines 211 – 213, referring to reference 52. At least I don’t know what the authors aim to inform with these lines!?
I suggest to replace “carcinogenic” (line 219) with “cancer”, or “tumor micro”.
Line 266; “They play a direct role in inducing tumor cell death and….”. This bold statement would need a reference.
Please also clarify lines 221 – 225.
Please rewrite line 237 “the suppression of antitumor T cells, including CD8+ T and NK cells”. NK cells are not T cells, but here they are given as an example of antitumor T cells.
I guess “aerobia” (line 448) should be “aerobic”
Comments on the Quality of English Languageminor revision of language required
Author Response
RE: ijms-3135030
Dear Ms. Hyacinth Hong,
We would like to thank the reviewers for their thoughtful comments regarding our work. They have definitely helped to improve our manuscript. In this manuscript version, we amended some parts to improve clarity and precision, as suggested. All of our changes are highlighted in red. In brief, the main changes in the manuscript were the following:
- We made the writing changes suggested by reviewer #1.
- We clarified some sentences.
- We added the requested references.
- We improved the quality of the tables.
- We added a paragraph about the Warburg effect and exercise as suggested by reviewer #2.
- We corrected figure 2B.
We look forward to hearing from you soon regarding the status of our manuscript.
Yours sincerely
Miguel Conceição, Ph.D.
REVIEWER #1:
Comments and Suggestions for Authors
The manuscript by Buzaglo et al., “The therapeutic potential of physical exercise in cancer: the role of chemokines”, is a review focused on the actions of chemokines and their role – positive and negative – in governing the interactions between the immune system and cancer. As reflected in the title, with an emphasis on how exercise influence chemokine axes, and how that could be exploited in cancer therapies.
The review is very timely, organized and written. Below some points raised.
I raise the following concerns.
The authors have chosen the term “chronic” for repeated bouts of (acute) exercise. I am not sure that is the most appropriate term. I think it makes more sense to speak about repeated (for whatever period relevant) bouts or regular (programs of) acute exercise. Chronic exercise, e.g., running for really long time, e.g., marathon and beyond could perhaps signify as chronic, but would not be likely to set a similar anti-tumor state of the immune system?!?
Response: Thank you for your valuable feedback. The reviewer raised a valid point. In the revised version of the manuscript, we have replaced the term “chronic” with “repeated exercise sessions” where appropriate. However, we continue to use “chronic” when specifically referring to the immune system’s long-term adaptations resulting from repeated exercise sessions, which is a common term in the literature.
The term “tumor behaviour” (first used line 48) is odd, I think. Would “tumor composition” or “cellular and molecular composition of the tumor” be more appropriate?
Response: Thank you for your observation. The term “tumor behaviour” is widely used in the literature. A quick search on PubMed yields over two thousand articles using this term (https://pubmed.ncbi.nlm.nih.gov/?term=%22tumor+behavior%22&sort=pubdate – Please see the figure below). Given its extensive usage in scientific research, we have chosen to retain the term in the manuscript.
The paragraph starting line 258, is set of in the first line as if it will be about Th1 cells, but in fact its more about B cells.
Response: Thank you for your insightful feedback. The purpose of the paragraph is to highlight the involvement of immune cells, such as Th1 and B cells, in the antitumor response. In the revised version of the manuscript, we have included additional information on Th1 cells to better balance the content concerning B cells (see lines 259-261). Additionally, we have revised several sentences to enhance clarity (see lines 263-267).
The review is mainly focused on data from mouse tumor models, reflecting that the research field of exercise immunology/oncology is understudied in the human setting. Nonetheless, some data are available that could potentially be used to strengthen the notion that the mouse studies can be translated to the human setting. Also, clinical studies are in progress combining exercise with immunotherapy in cancer, coupled with biological monitoring (PMID: 35247994).
Response: Thank you once again for your valuable feedback. The reviewer raises a valid point. However, the current literature includes only a limited number of studies conducted in humans that specifically investigate the mechanisms of action of chemokines and their action in the tumor following physical exercise. As a result, our discussion includes more studies using animal models than human models. In this revised version of the manuscript, we have highlighted that only a few studies that explored the chemokine-exercise-tumor relationship in humans and emphasized the need for more translational research in this area (Lines 533-537).
Regarding the combination of immunotherapy, exercise, and cancer, we understand that further discussion is premature at this stage, as ongoing research has yet to yield sufficient data. We believe it is important to await more robust findings before forming well-supported hypotheses.
Minor concerns
At least on my print and screen the quality/resolution of fig. 1 is low.
Response: The figure now has 300 pixels per inch (PPI), ensuring accurate colors and sharp details.
Several place there is a lack of space between last letter and reference, i.e., “…..immune[“, which should be “….immune [“
Response: Thank you for the feedback. This was changed accordingly.
When authors are referred to in the text it should be e.g., “Pedersen et al.” and not “Pedersen L. et al.”)
Response: It was accordingly changed.
I suggest to replace “induction” (line 86) with “mobilization”.
Response: Thank you for the suggestion. Really the word “mobilization” fits better in this sentence (line 85).
I suggest to replace “immunological cells” (line 89) with “immune cells”.
Response: It was accordingly changed (line 86).
I suggest to replace “induced” (line 89) with “attracted”.
Response: It was accordingly changed (line 87).
I suggest to clarify the lines 211 - 213, referring to reference 52. At least I do not know what the authors intend to inform with these lines!?
Response: In this manuscript version, we rewrite the sentence to improve clarity. Please, see lines 207-214.
I suggest to replace “carcinogenic” (line 219) with “cancer”, or “tumor micro”.
Response: It was accordingly changed (line 221)
Line 266; “They play a direct role in inducing tumor cell death and….”. This bold statement would need a reference.
Response: We added references as suggested (271).
“ 71. Bindea, G.; Mlecnik, B.; Tosolini, M.; Kirilovsky, A.; Waldner, M.; Obenauf, A.C.; Angell, H.; Fredriksen, T.; Lafontaine, L.; Berger, A.; et al. Spatiotemporal Dynamics of Intratumoral Immune Cells Reveal the Immune Landscape in Human Cancer. Immunity 2013, 39, 782–795, doi: 10.1016/j.immuni.2013.10.003.
- Haabeth, O.A.W.; Lorvik, K.B.; Hammarström, C.; Donaldson, I.M.; Haraldsen, G.; Bogen, B.; Corthay, A. Inflammation Driven by Tumour-Specific Th1 Cells Protects against B-Cell Cancer. Nat Commun 2011, 2, 240, doi:10.1038/ncomms1239.
- Kinker, G.S.; Vitiello, G.A.F.; Ferreira, W.A.S.; Chaves, A.S.; Cordeiro de Lima, V.C.; Medina, T. da S. B Cell Orchestration of Anti-Tumor Immune Responses: A Matter of Cell Localization and Communication. Front Cell Dev Biol 2021, 9, doi:10.3389/fcell.2021.678127.”
Please also clarify lines 221 – 225.
Response: In this manuscript version, we rewrite the sentence to improve clarity. Please, see lines 217-220.
Please rewrite line 237 “the suppression of antitumor T cells, including CD8+ T and NK cells”. NK cells are not T cells, but here they are given as an example of antitumor T cells.
Response: It was accordingly changed (Line 237-239).
I guess “aerobia” (line 448) should be “aerobic”
Response: It was accordingly changed (Line 449).
Comments on the Quality of English Language minor revision of language required
Response: The manuscript was revised by a native English speaker.
REVIEWER #2:
Comments and Suggestions for Authors
In their review, Glenda et al. examine the critical role of chemokines in cancer progression and the potential therapeutic impact of physical exercise. The review highlights how physical exercise, particularly aerobic and resistance training, can modulate chemokine expression, fostering a more favorable inflammatory profile in cancer patients. This modulation improves intratumoral vasculature and boosts the infiltration of antitumor immune cells, creating an environment less supportive of tumor growth. Glenda et al. suggest that leveraging exercise-induced chemokine modulation could be a promising strategy for enhancing cancer treatment and prevention.
Response: We Thank the reviewer for the valuable comments.
Major Concerns
One aspect the authors could further explore is the application of physical exercise as an adjunct to cancer immunotherapy. Specifically, the Warburg effect—a metabolic shift in cancer cells that leads to increased lactic acid production and subsequently raises the acidity of the tumor microenvironment (TME)—has been recognized as a significant therapeutic target. This acidic environment can suppress immune cell function and promote tumor progression. However, physical exercise, particularly aerobic exercise, has the potential to influence this process by improving oxygenation and reducing lactic acid accumulation in the TME. This could, in turn, enhance the efficacy of immunotherapies by creating a less hostile environment for immune cells.
On the other hand, certain types of resistance training could temporarily increase lactic acid levels in muscles. It would be valuable to investigate whether this increase could inadvertently affect the TME's acidity and potentially impact the effectiveness of cancer treatments. Understanding the nuanced effects of different exercise regimens on the TME could provide critical insights into how best to integrate physical activity into cancer treatment protocols.
Response:
We appreciate the suggestion and have added the following paragraph linking physical exercise with the Warburg effect and the consequent immune response (lines 491-512):
Linked to immune suppression triggered by hypoxia, lactate metabolism in the TME is also relevant [120–122]. An important characteristic of tumors is the Warburg effect, a phenomenon described by Otto Warburg in which cancer cells prefer to metabolize glucose through anaerobic glycolysis (i.e. lactic acid fermentation) instead of producing energy through oxidative phosphorylation [120]. The tumor glycolytic phenotype increases lactate production by cancer cells, making TME more acidic and inhibiting T cell responses, which may promote the production and greater circulation of pro-inflammatory chemokines, damaging the immune pathways that could regulate tumor growth [120]. So, several authors pointed out that lactate metabolism is determinant for malignancy, immune suppression, and cancer prognosis [120,121,123]. Since increased glucose consumption is directly linked to tumorigenesis, reversing this metabolic profile could compromise tumor growth and survival [120]. Importantly, regular physical exercise, especially high-intensity exercise, can impact tumor metabolism, inhibiting glycolysis [120,123]. Parolin et al. [124] analysed the time course for activating glycogen phosphorylase and pyruvate dehydrogenase in human skeletal muscle during repeated bouts of maximal exercise. The authors demonstrated that with successive bouts, the ability to stimulate substrate phosphorylation through phosphocreatine hydrolysis and glycolysis decreased, shifting metabolism toward greater dependence on oxidative phosphorylation [124]. In summary, the induced acidosis during exercise leads to reduced lactate production as a result of glycogenolysis inhibition [125]. These lactate-inhibiting effects may enhance the distribution and transmigration of immune cells and anti-tumor chemokines to tumors, facilitating tumor lysis [126,127].
Minor comments:
Format of table is off.
Response: Thank you for the feedback. Tables 1 and 2 have been updated and inserted in the file as text.
Typo in Fig2B. Lymphocytosis
Response: Thank you for the feedback. Fig.2B is updated with the correction requested.

Reviewer 2 Report
Comments and Suggestions for Authors
In their review, Glenda et al. examine the critical role of chemokines in cancer progression and the potential therapeutic impact of physical exercise. The review highlights how physical exercise, particularly aerobic and resistance training, can modulate chemokine expression, fostering a more favorable inflammatory profile in cancer patients. This modulation improves intratumoral vasculature and boosts the infiltration of antitumor immune cells, creating an environment less supportive of tumor growth. Glenda et al. suggest that leveraging exercise-induced chemokine modulation could be a promising strategy for enhancing cancer treatment and prevention.
Major comments:
One aspect the authors could further explore is the application of physical exercise as an adjunct to cancer immunotherapy. Specifically, the Warburg effect—a metabolic shift in cancer cells that leads to increased lactic acid production and subsequently raises the acidity of the tumor microenvironment (TME)—has been recognized as a significant therapeutic target. This acidic environment can suppress immune cell function and promote tumor progression. However, physical exercise, particularly aerobic exercise, has the potential to influence this process by improving oxygenation and reducing lactic acid accumulation in the TME. This could, in turn, enhance the efficacy of immunotherapies by creating a less hostile environment for immune cells.
On the other hand, certain types of resistance training could temporarily increase lactic acid levels in muscles. It would be valuable to investigate whether this increase could inadvertently affect the TME's acidity and potentially impact the effectiveness of cancer treatments. Understanding the nuanced effects of different exercise regimens on the TME could provide critical insights into how best to integrate physical activity into cancer treatment protocols.
Minor comments:
1. Format of table is off.
2. Typo in Fig2B. Lymphocytosis
Comments on the Quality of English Languagena
Author Response

(The authors gave the same response as above.)
